# Identification of CSF3R Mutations in B-Lineage Acute Lymphoblastic Leukemia Using Comprehensive Cancer Panel and Next-Generation Sequencing

**DOI:** 10.3390/genes12091326

**Published:** 2021-08-27

**Authors:** Mamoon Rashid, Abdulrahman Alasiri, Mohammad A. Al Balwi, Aziza Alkhaldi, Ahmed Alsuhaibani, Abdulrahman Alsultan, Talal Alharbi, Lamya Alomair, Bader Almuzzaini

**Affiliations:** 1King Abdullah International Medical Research Center (KAIMRC), Department of Bionformatics, Ministry of National Guard Health Affairs, King Saud Bin Abdulaziz University for Health Sciences (KSAU-HS), Riyadh 11426, Saudi Arabia; rashidma@ngha.med.sa (M.R.); omairl@ngha.med.sa (L.A.); 2King Abdullah International Medical Research Center (KAIMRC), Medical Genomics Research Department, Ministry of National Guard Health Affairs, King Saud Bin Abdulaziz University for Health Sciences (KSAU-HS), Riyadh 11426, Saudi Arabia; ai7alasiri@gmail.com (A.A.); balwim@ngha.med.sa (M.A.A.B.); 3Pathology and Laboratory Medicine, King Abdul Aziz Medical City, Ministry of National Guard Health Affairs, King Saud Bin Abdul Aziz University for Health Sciences (KSAU-HS), Riyadh 11426, Saudi Arabia; khaldia@ngha.med.sa (A.A.); suhaiban@hotmail.com (A.A.); 4Department of Pediatric Hematology/Oncology, King Abdullah Specialist Children’s Hospital, Ministry of National Guard Health Affairs, King Saud Bin Abdul Aziz University for Health Sciences (KSAU-HS), Riyadh 11426, Saudi Arabia; aalsultan@gmail.com (A.A.); harbit@ngha.med.sa (T.A.)

**Keywords:** leukemia, acute lymphoblastic leukemia, B-ALL, mutations, *CSF3R*, cancer

## Abstract

B-lineage acute lymphocytic leukemia (B-ALL) is characterized by different genetic aberrations at a chromosomal and gene level which are very crucial for diagnosis, prognosis and risk assessment of the disease. However, there is still controversial arguments in regard to disease outcomes in specific genetic abnormalities, e.g., 9p-deletion. Moreover, in absence of cytogenetic abnormalities it is difficult to predict B-ALL progression. Here, we use the advantage of Next-generation sequencing (NGS) technology to study the mutation landscape of 12 patients with B-ALL using Comprehensive Cancer Panel (CCP) which covers the most common mutated cancer genes. Our results describe new mutations in *CSF3R* gene including S661N, S557G, and Q170X which might be associated with disease progression.

## 1. Introduction

B-lineage acute lymphoblastic leukemia (B-ALL) is the most common type of childhood malignancy. It originates from B-cell progenitors in the bone marrow [1]. Risk assessment and classification of B-ALL is based on different criteria including age, white blood cell (WBC) count, central nervous system (CNS) involvement, and chromosomal aberration. Cytogenetic and molecular assessment of B-ALL provides significant insight into risk stratification, treatment response and prognosis. Conventional karyotyping, fluorescence in situ hybridization (FISH), comparative genomic hybridization (CGH), and molecular methods such as polymerase chain reaction (PCR) or reverse transcript quantitative PCR (RT-qPCR) are used for molecular characterization of leukemia [2,3,4,5,6,7,8]. *ETV6–RUNX1* rearrangement is the most common genetic subtype of childhood B-ALL and is associated with good prognosis. However, other subtypes such as *BCR–ABL1*, *BCR–ABL1* like, *KMT2A* rearrangement, and hypodiploidy are associated with a less favorable outcome [6,9,10].

The impact of chromosome 9p deletion as an independent adverse prognostic marker in B-ALL is debatable. Chromosome 9p abnormalities were associated with a higher risk of treatment failure in childhood B-ALL in an early report by a children’s cancer group and also in adult B-ALL [11]. However, MRC UKALL X in childhood B-ALL did not show prognostic value of del (9p) [12]. To the contrary, MRC UKALLXII/ECOG 2993 trial in adult ALL showed an association between del (9p) and improved outcomes [13]. The variability between these studies could be explained by co-existing cytogenetic abnormalities or the need to take into consideration the minimal residual disease (MRD) assessment early in the treatment course. There are cases with normal karyotypes but still with a relapsed condition.

In this study we are reporting on B-ALL patients, few of which are normal cytogenetically but still showed relapsed. We utilized the Ion AmpliSeq Comprehensive Cancer Panel (CCP) target panel which covers 409 oncogenes and tumor suppressor genes to define the mutational landscape of 12 pediatric patients with B-ALL.

## 2. Materials and Methods

### 2.1. Patients

Bone marrow (BM) aspirates from 12 children diagnosed with B-ALL (see Table 1) were collected at the pediatric hematology/oncology department at King Abdullah Specialized Children’s hospital and King Abdulaziz Medical City-Riyadh after getting informed consent for genetic studies and after ethical approval from our IRB (RC-16-157-R).

### 2.2. Karyotyping and FISH

Conventional karyotyping and FISH were performed for all of the patients using sodium heparinized bone marrow samples following classical protocol. FISH was done for B-ALL panel probes to assess the presence and absence of *BCR/ABL*, *PML/RARA*, *TEL/AML*, P16, 9Pdeletion, all used probes from VYSIS and process according to the manufacturer’s protocol.

### 2.3. DNA Isolation and Target Sequencing

DNA was extracted from bone marrow samples using a QIAampDNA mini kit from QIAGEN according to the manufacturer’s protocol. DNA quantity and integrity were evaluated using NanoDrop 2000 (Thermo-fisher Scientific, Waltham, MA, USA) and Qubit^®^ 3.0 Fluorometer (Thermo-fisher Scientific) following the manufacturer’s protocol. An Ion Torrent adapter-ligated library was generated following the manufacturer’s protocol (Ion AmpliSeq™ CCP kit PIv3, Rev. A.0; MAN0010084; Thermo Fisher Scientific, Inc.). Briefly, 100 ng high-quality genomic DNA was used to construct a library using the Ion AmpliSeq™ Library Kit 2.0. Pooled amplicons were end-repaired, and Ion Torrent adapters and amplicons were ligated with DNA ligase. Following AMPure bead purification (Beckman Coulter, Inc., Brea, CA, USA), the concentration and size of the library were determined using the Applied Biosystems^®^ StepOne™ Real-Time PCR system and Ion Library TaqMan^®^ Quantitation kit (both from Thermo Fisher Scientific, Inc.). Sample emulsion PCR, emulsion breaking, and enrichment were performed using the Ion PI™ Hi-Q™ Chef 200 kit (Thermo Fisher Scientific, Inc.), according to the manufacturer’s instructions. An input concentration of one DNA template copy per ion sphere particle (ISPs) was added to the emulsion PCR master mix and the emulsion was generated using the Ion Chef™ System (Thermo Fisher Scientific, Inc.). Template-positive ISPs were enriched, sequencing was performed using Ion PI™ Chip kit v3 chips on the Ion Torrent Proton, and barcoding was performed using the Ion DNA Barcoding kit (Thermo Fisher Scientific, Inc.).

### 2.4. Comprehensive Cancer Panel

A pre-designed comprehensive cancer panel (CCP) from Ion AmpliSeq™ (Life Technologies, Carlsbad, CA, USA) was used, with four pools of primers comprised of 16,000 primer pairs of 409 genes that cover 15,749 somatic mutations in the catalogue of somatic mutation (COSMIC) database.

### 2.5. Sequence Alignment and Variant Calling

Data from Ion Proton were processed using Torrent Suite Software (v5.0.2) for Bioinformatics base calling, removal of low-quality reads, adapter trimming, and alignment against the human reference genome hg19 build (GRCh37-hg19) using the Torrent Mapping Alignment Program (TMAP). Variant calling was conducted using the Torrent Variant Caller (TVC) plugin (v5.0). Read mapping and the aligned sequence was automatically transferred into the Ion Reporter Server (v5.0) to identify SNV (Single Nucleotide Variation), MNV (Multi-nucleotide Variation), and InDel (Insertion Deletion).

### 2.6. Variant Annotation

Web ANNOVAR (wANNOVAR) [14] was used for annotation of the variants including gene-based, region-based and filter-based annotations on a variant call format (VCF) file generated from Ion Reporter Server.

### 2.7. Exploration of B-ALL Mutation Data Using Maftools

Analysis of a large volume of cancer genomics data needs numerous independent statistical and computational tools to identify driver gene(s), pathways, enrichments, gene signatures, and associated prognostic factors of the cancer under study. Maftools [15], an R package, provides a one-stop-shop, facilitating all such analyses on the large-scale cancer genomics dataset. In this manuscript we used Maftools to analyze somatic mutations (SNV, MNV, and InDels) identified by panel sequencing of 12 B-ALL patients. Maftools require variants in the Mutation Annotation Format (MAF). Since annotations were conducted using wANNOVAR, the output format was not compatible to Maftools. Therefore, the “annovarToMaf” function (in-built in Maftools) was used to convert the annovar output table file into a MAF file. MAF file was then imported by the “read.maf” function to obtain a MAF object compatible for all downstream analyses by Maftools.

## 3. Results

A comprehensive cancer panel was used to evaluate somatic mutations in 12 children with B-ALL from King Abdulaziz Medical City (KAMC); clinical characteristics are shown in Table 1. An average of 9,736,469 reads were generated for each sample with a mean coverage depth of 464.8752. The total number of mutations in 12 samples is 32,179. However, the unique number of mutations across all the samples came to be 11,007. We filtered out the variants based on allele frequency from the 1000 genome dataset; 22,373 variants were below or equal to the allele frequency of 0.1. The rationale behind this filtering was to enrich the variants for somatic mutations, i.e., polymorphism was removed from our dataset. The unique number of mutations turned out to be 9085 across all samples.

### 3.1. Mutational Landscape of B-ALL

We analyzed the variant data using “Maftools”. We used 22,373 variants set to process further with Maftools. During preprocessing, 4541 and 6765 (total 11,306) variants were removed as duplicated and silent variants, respectively. Therefore, only 11,067 somatic variants (22,373–11,306) were considered by Maftools for further analysis and reporting. Figure 1 illustrates the summary of genetic variants found in 12 B-ALL patients.

Summary: Figure 1 describes the general features of the mutations across the B-ALL patients; variant classification shows the most abundant is missense mutations. The single nucleotide variant (SNV) is the most abundant type. Among the SNV class, transitions are more abundant than transversions. Median number of variants per sample is ~320. We observed that three samples have very high mutational load. The bottom right panel in Figure 1 shows the barplot of top 10 genes in our patient cohort with high number of mutations. Each bar represents the number of mutations (on x-axis) for each gene and the stacked color indicates variant classification. The number on top of the bar represents the percentage of samples with mutation(s) in that particular gene.

The oncoplot in Figure 2 represents the top 20 mutated genes across the 12 samples. An oncoplot is a combination of gene level summary, sample level summary, and the patient clinical information providing substrates for the selection of potential oncogenes in the cancer cohort. The most frequently mutated genes in our B-ALL patients were *PDE4DIP, TET2, KMT2C, CBL, NOTCH4, FGFR3, KMT2D, SYNE1, NIN*, and *CHEK1*. Excluding the *SYNE1*, *NOTCH4*, and *CHEK1* genes, all others are well known leukemia/lymphoma cancer genes with a tier 1 category in the Cancer Gene Census database. Therefore, 7/10 of the top frequently mutated genes in our B-ALL cohort are known cancer genes.

### 3.2. Comparison with TCGA Signature

We also compared the mutational load of the B-ALL samples with that of the other TCGA (The Cancer Genome Atlas) datasets (Figure 3). Our B-ALL dataset has high mutational load second largest to the skin cutaneous melanoma (SKMC), but close to the Diffuse Large B-cell Lymphoma (DLBC) in TCGA dataset. Therefore, the mutational load of our B-ALL dataset in is concordance with that of the TCGA dataset, with slight deviation.

### 3.3. Signature Pathways across B-ALL Samples

The pathway analysis revealed some signature pathways in the B-ALL patients studied in this manuscript. Among the top affected pathways were RTK-RAS, PI3K, NOTCH, Cell_cycle, TP53, etc. which are affected across a substantial number of B-ALL samples (Figure 4). The Ras pathway, Notch pathway, epigenetic modification, and cell-cycle regulation are already known for their involvement in childhood ALL [16]. Therefore, the most affected pathways discovered in our B-ALL cohort were in agreement with previous findings.

### 3.4. Somatic Interaction of Mutations

Mutually exclusive or co-occurring mutational events are very important in cancer biology. To analyze such events, a Pair-wise Fisher’s Exact test was used. TET2 with FGFR3 and NOTCH4, and NOTCH4 with FGFR3 showed significant co-occurring mutations (Figure 5). This interplay of genetic mutations in TET2, FGFR3, and NOTCH4 genes could potentially be indicative of B-ALL pathogenesis.

### 3.5. Potential Cancer Driver Genes in B-ALL

The gene(s), when mutated, provide selective growth advantages to tumor cells and are defined as cancer driver gene(s). These driver mutations/genes are positively selected during clonal evolution of tumors. To identify potential cancer driver genes we applied the algorithm called “oncodrive” within Maftools. Oncodrive is based on the OncodriveCLUST algorithm [17], which leverages the observation that most of the activating mutations within oncogenes are clustered around mutational hotspots. We could not find any cancer driver genes statistically significant in our B-ALL dataset (Figure 6). However, when we sorted the output of the oncodrive algorithm on the “number of mutations in clusters” we did see a lot of oncogenes on the top of the list (Appendix A). This list included TP53, APC, PTEN, EGFR, VHL, PDE4DIP, CDKN2A, TET2, PIK3CA, KIT, etc. Similarly, the list when sorted out on the “fraction of mutations in clusters” contained important oncogenes in leukemia CTNNB1, CEBPA, SOCS1, NRAS, IDH1, NPM1, PPP2R1A, PTPRT, MYD88, and IDH2. The result of the “oncodrive” turned out to be statistically insignificant due to very small number of samples (i.e., 12).

### 3.6. CSF3R Mutations

CSF3R’s recently emerging role in leukemia [18,19,20,21] prompted us to investigate the genetic alterations of this gene in our B-ALL cohort. Surprisingly, we found three nonsynonymous (S661N, S557G, D320N) and one stopgain (Q170X) mutation (Figure 7) in addition to various other non-exonic mutations in the CSF3R gene (Appendix A). Out of these four mutations, only one (D320N) is known in the dbSNP database with rs3918018 (probably a polymorphism), and the other three are novel somatic mutations. Moreover, the D320N is also reported to be a confirmed somatic mutation in the COSMIC database. These four potentially activating mutations occur exclusivity in four different B-ALL patients. S661N, S557G, and Q170X CSF3R mutations occurred in patients 2106, 2110 and 2112, respectively, who showed relapses, while patients bearing the D320N mutation showed remission (Figure 2). These three patients (2106, 2110, 2112) were normal in the cytogenetic and FISH assay without 9-p deletion. Therefore, we hypothesize that these nonsynonymous potential activating CSF3R mutations could explain the pathogenesis of a few of the B-cell acute lymphoblastic leukemia patients. Membrane proximal mutations (such as S661N and S557G in this current study) in CSF3R were reported to operate through the JAK/STAT signaling pathway and truncation mutation (such as Q170X) through SFK-TNK2 signaling pathways [21].

Genetic alterations in the CSF3R gene have been reported recently in different leukemias of myeloid and lymphoid origin. In Chronic Neutrophilic Leukemia (CNL) and atypical Chronic Myeloid Leukemia (CML), many CSF3R oncogenic mutations have been reported to segregate to two distinct regions of CSF3R, leading to preferential downstream signaling pathways through the SRC family–TNK2 or JAK kinases [21]. A patient with CNL carrying the JAK-activating CSF3R mutation improved significantly after administration of JAK1/2 inhibitor ruxolitinib. Large-scale studies to calculate the incidence of CSF3R mutations in AML and ALL showed that these are not common mutations but are often associated with genetic alterations in core-binding factor gene abnormalities [19]. Prognostic impacts of CSF3R mutations have been studied in a large number of patients (>2000) with pediatric AML, and have shown that CSF3R truncation mutation is rare in pediatric AML [18]. Very recently, a truncation mutation in CSF3R in a patient with B-cell acute lymphoblastic leukemia was reported, and the patient was shown to have a favorable response to chemotherapy plus dasatinib [20]. This study is supportive of our finding of a truncation mutation Q170X in the CSF3R gene in B-ALL patient. Conclusively, we report novel and known genetic variants of CSF3R in patients with B-ALL in our study. The functional implications of these CSF3R mutations require further characterization of a large number of B-ALL patients and their samples.

## 4. Conclusions

CCP resulted in a mutational landscape of B-ALL patients with 9-p deletion and/or negative cytogenetic or FISH examination. The analysis of these mutational landscapes identified both novel and known somatic mutations in known cancer genes and also in novel genes such as CSF3R in B-ALL patients. We observed two types of mutations in the CSF3R gene (reported in literature): the first type of mutations lie in membrane proximal regions (D320N, S557G, S661N) and the second type is a truncation or nonsense mutation (Q170X). Although Q170X leads to a truncated protein product, further functional validation is required to understand its mechanism of activation of the downstream signaling pathway. Mutations in the CSF3R gene could potentially explain the relapse of the three patients with normal cytogenetic and FISH results. Therefore, we conclude that the cytokine receptor family, e.g., CSF3R, should be explored for its oncogenic potential in B-ALL.

Moreover, this study has some inherent limitations, such as a small sample size, lack of functional validation and characterization, etc. The functional validation of identified somatic mutations in CSF3R would be very useful, and we are working along this line. Availability of large number of samples under a multi-center program/consortium to overcome the difficulty of sample recruitment would lead to validation of our findings and association studies with the clinical data.

## Figures and Tables

**Figure 1 genes-12-01326-f001:**
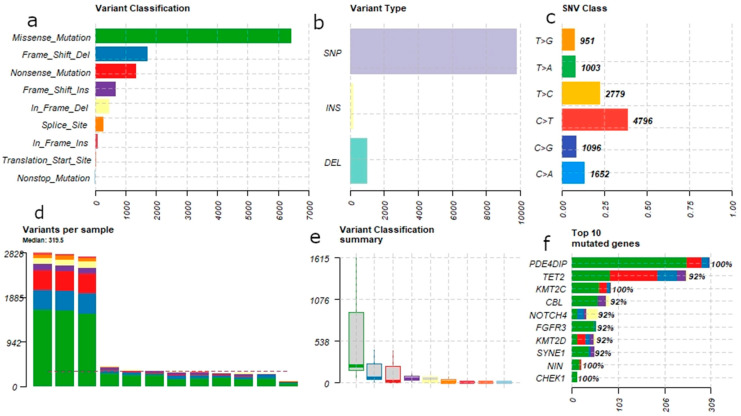
Summary of somatic mutations in 12 B-ALL patients. Six panels above describe the variant classification, variant type, SNV class, variant per sample, variant classification summary and top 10 mutated genes. Missense mutations are the most common form of mutations (**a**). Single nucleotide polymorphism (SNP) is the most common genetic variant type (**b**). Among single nucleotide variation (SNV) transition, C > T is the most common (**c**). (**d**) represents the mutational load per sample. The variant classification summary (**e**) represents box plot of the numbers across the 12 B-ALL patients. At the end, the most frequently mutated genes have been shown across the cohort (**f**).

**Figure 2 genes-12-01326-f002:**
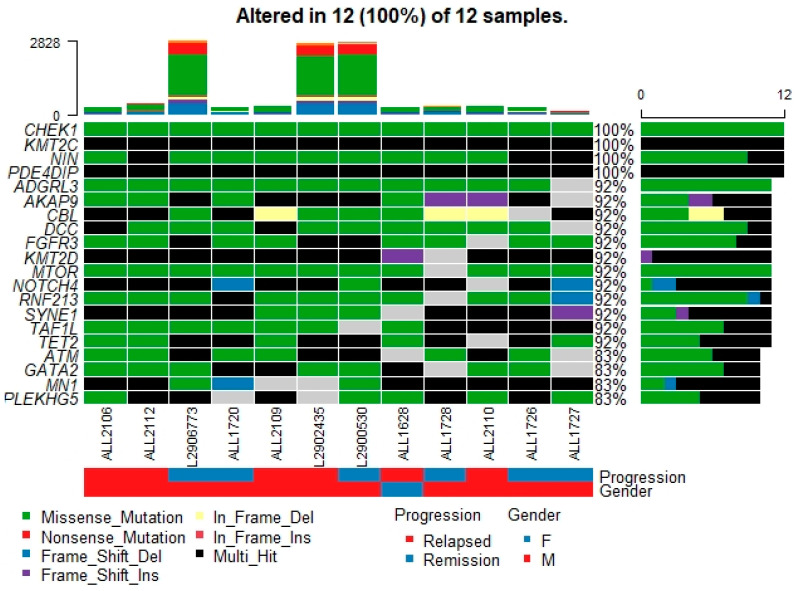
An oncoplot showing the heatmap of variant classification for the top 20 genes across 12 B-ALL samples. The X-axis shows different samples and the Y-axis shows thetop 20 genes. CHEK1, KMT2C, NIN, and PDE4DIP genes harbor mutations across all tumor samples (100%) showing their importance in oncogenesis.

**Figure 3 genes-12-01326-f003:**
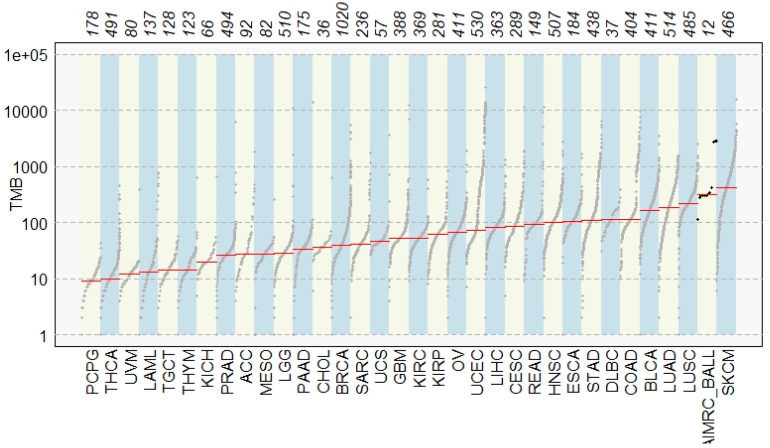
Comparison of mutational load with TCGA datasets. The Y-axis shows the number of mutations per mega base pair of the genomic sequence. The mutational load of the KAIMRC_BALL dataset from this study is the second largest, showing similarity with Skin Cutaneous Melanoma (SKCM), Lung squamous cell carcinoma (LUSC), Lung adenocarcinoma (LUAD), Bladder Urothelial Carcinoma (BLCA), Colon adenocarcinoma (COAD), and most importantly with Diffuse Large B-cell Lymphoma (DLBC).

**Figure 4 genes-12-01326-f004:**
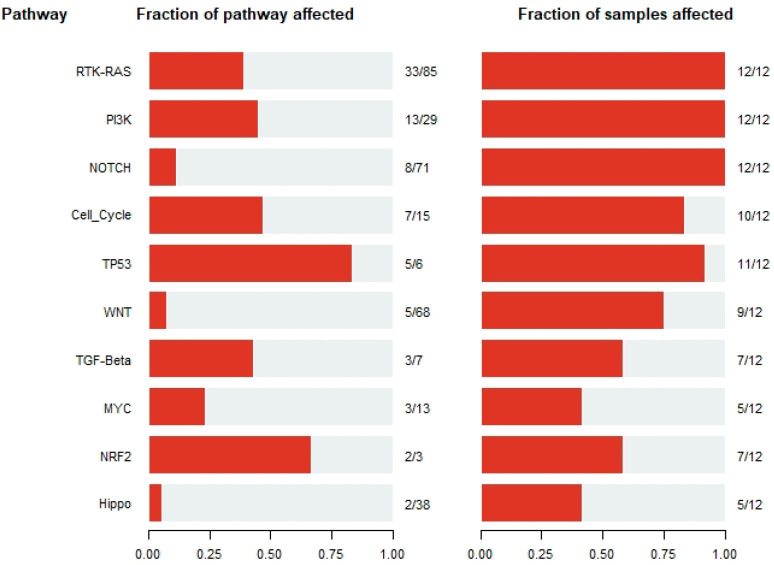
Most affected pathways across B-ALL samples.

**Figure 5 genes-12-01326-f005:**
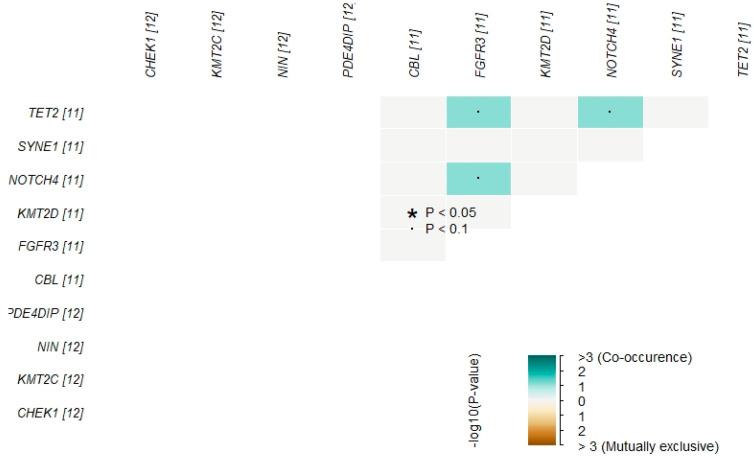
Pair-wise Fisher’s Exact test analysis of top 10 highly mutated genes to detect mutually exclusive or co-occurring events. * *p* < 0.05, ^·^
*p* < 0.1.

**Figure 6 genes-12-01326-f006:**
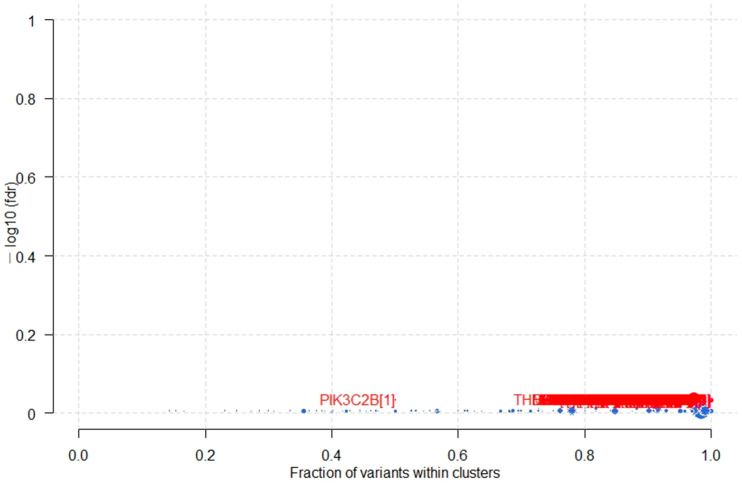
B-ALL associated potential driver genes identified by oncodrive in 12 B-ALL patients. The results are not statistically significant probably due to a smaller number of samples in the B-ALL cohort.

**Figure 7 genes-12-01326-f007:**
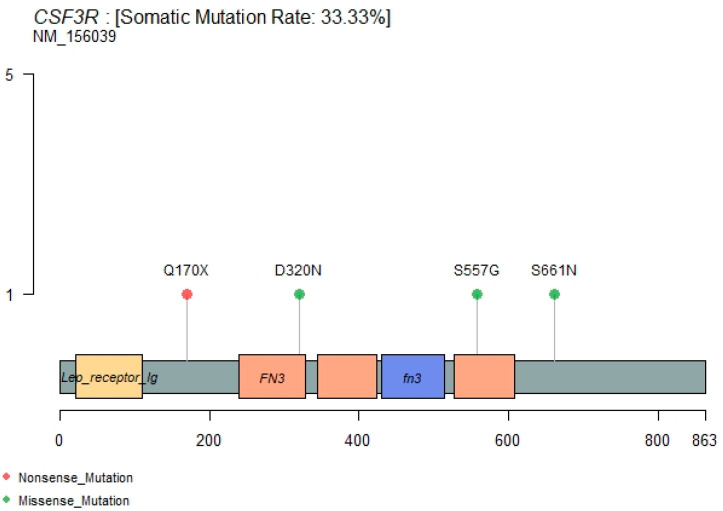
Lollipop plot for CSF3R mutations.

**Table 1 genes-12-01326-t001:** Clinical Characteristics of the 12 B-ALL Patients Studied in this Manuscript.

Case	Age at Diagnosis (Years)/Sex	WBC	Blast	Cytogenetic	FISH	Progression
1628	4/F	27.1	32%	46,XX,i(22)(?q10)(3)/46,XX(19)	nuc ish(P16x0,9cenx2), (BCRx4,ABL1x2)(61/200)/(BCRx3,ABL1x2)(37/200), (P53x2,D17Z1x3)(34/200)/(P53,D17Z1)x3(20/200)	Relapse
1720	6/M	1.4	60%	NA	Nuc ish(p16x0,9cenx2,D17Z1x3)(17/200)	Remission
1726	12/M	6.3	30%	44-46,XY,del(2)(p?22),?dic(9;17)(p?13;p?11.2),?der(14;17)(?p11.2;?p13),+mar,inc(cp3)	9p deletion	Remission
1727	3/M	10.5	85%	46,XY,del(11)(q13q23),inc(19)/46,XY (2)	9p, MLL deletion and TEL/AML fusion	Remission
1728	8/M	9.5	13%	52-55,XY,+X,+X,+4,+6,+8,dic(9;17)t(?p22;q10),+14,+18,+21,+21,inc(cp12)/46,XY(5)	9p deletion and hyperdiploidy	Remission
2106 *	8/M	N.P	80%	46,XY(18)	NA	Relapsed
2109	8/M	5.2	3%	46,XY,del(9)(p21)(8)/46,XY(12)	9p deletion	Relapsed
2110 *	10/M	7.9	80%	46,XY(20)	TEL/AML positive	Relapsed
2112 *	3/M	16.5	84%	46,XY(20)	Negative	Relapsed
2902435	11/M	1.09	4%	46,XY(18)	9p21,extra copy of aml1	Relapsed
2906773	15/M	10	19%	46,XY,del(9)(q21),del(13)(q21)(17)	P16 gene deletion	Remission
2900530	20/M	2.8	30%	NA	P16 gene deletion	Remission

* These cases are bearing CSF3R mutations; NA, not available.

## Data Availability

Data will be available on request.

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
