# Peer review of "Identification of CSF3R Mutations in B-Lineage Acute Lymphoblastic Leukemia Using Comprehensive Cancer Panel and Next-Generation Sequencing"

_genes, 2021, doi:10.3390/genes12091326_

Round 1

Reviewer 1 Report

The article entitled “Identification of CSF3R mutations in B-lineage acute lymphoblastic leukemia using comprehensive cancer panel and next-generation sequencing “by M. Rashid et al. reported identification of mutation in B-lineage ALL.

The article is interesting and contributes to the trend of a new classification of leukemia according to several parameters including the mutation profile.

The article in general is well presented but several points should be taking into account.

  • Usually, children are under 15 years. In the group of patients, one is 15 and another is 20.
  • In figure 1 the panels are probably excessive and not easily readable.
  • In figure 2 the significance of yellow is not described.
  • The paragraph 3.6 CSF3R mutation can be written in a more comprehensible way and leads to a clear conclusion according to the type of mutation.
  • Figure 6 should be corrected.

Author Response

Q: The article entitled “Identification of CSF3R mutations in B-lineage acute lymphoblastic leukemia using comprehensive cancer panel and next-generation sequencing “by M. Rashid et al. reported identification of mutation in B-lineage ALL.

A: Thank you to reviewer to review our manuscript in this time of pandemic.

Q: The article is interesting and contributes to the trend of a new classification of leukemia according to several parameters including the mutation profile.

A: We are thankful to reviewer for recognizing the potential of this manuscript.

Q: The article in general is well presented but several points should be taking into account.

A: We are happy to address those points raised by the reviewer.

Q: Usually, children are under 15 years. In the group of patients, one is 15 and another is 20.

A: We agree that we included a B-ALL patient of age 20 years. As per the United Nations convention the child should be defined as a human being below 18 years old. Since our study sample size is small so we intended to include this young patient in our study.

Q: In figure 1 the panels are probably excessive and not easily readable.

A: We are agreeing to the reviewer. Therefore, we elaborated the figure legend in order to understand the panels of the figure 1. We track changed the added text in the manuscript starting from line number 151.

Q: In figure 2 the significance of yellow is not described.

A: The yellow color code is for in frame deletion mentioned as In_Frame_Del below figure 2. Please see the manuscript.

Q: The paragraph 3.6 CSF3R mutation can be written in a more comprehensible way and leads to a clear conclusion according to the type of mutation.

A: We understand the reviewers concern and therefore added a sentence to paragraph 3.6 at line number 237 to state that two different types of mutations exist in CSF3R gene. These two types of mutations operate through different signaling pathways. The similar text also exist in the manuscript at line numbers 245-249.

Q: Figure 6 should be corrected.

A: We tried to make it clearer but due to high FDR values the points seem to fall on X-axis.

Reviewer 2 Report

The authors present NGS sequencing results on 12 pediatric patients with ALL. The paper describes in depth the techniques and mutational findings. The results mimic what has already been shown in ALL but with a very small sample size. No clinical correlation or functional data is provided.  The authors further discuss novel "mutations" in the CSF3R. The variant allele frequency is not provided and germline DNA was not sequenced to assess whether these single nucleotide changes were polymorphisms. The CSF3R missense mutations described are also not in areas of the protein previously shown to be associated with gain of function of the CSF3R. No functional studies are performed to determine if these are activating mutations or not. IN addition the third mutation is non-sense near the N-terminal of CSF3R and would be expected to create a non-functional protein. Again no functional studies to determine relevance of this "mutation".As tumor relevant CSF3R mutations cause activation of signaling, it is unclear what non-functional protein would have any role in leukemogenesis.  

Author Response

Q: The authors present NGS sequencing results on 12 pediatric patients with ALL. The paper describes in depth the techniques and mutational findings. The results mimic what has already been shown in ALL but with a very small sample size.

A: Thank you for reviewing our manuscript. We do accept that the sample size is small but the message to the Leukemia researcher or cancer biologist is significantly important. The importance of CSF3R (a cytokine receptor) as an oncogene is being recognized in leukemic patient and our manuscript is an early example. We are planning to perform large-scale sequencing with a decent sample size.

Q: No clinical correlation or functional data is provided.

A: Due to its small sample size, we could not perform correlation/association studies. Functional experiments are the part of the next bigger project we are conceiving.

Q: The authors further discuss novel "mutations" in the CSF3R. The variant allele frequency is not provided and germline DNA was not sequenced to assess whether these single nucleotide changes were polymorphisms.

A: The reviewer is correct. We focused on CSF3R due to the increasing number of evidences supporting its oncogenic potential in Leukemia patients.

Q: The CSF3R missense mutations described are also not in areas of the protein previously shown to be associated with gain of function of the CSF3R.

A: The transforming mutations in CSF3R shown previously include membrane proximal mutations such as T615A and T618I (1). The mutations of CSF3R described in our study (Q170X, D320N, S557G, S661N) are also membrane proximal mutations. Except the truncation mutation Q170X, the rest three mutations could be analogous to T615A or T618I.

Q: No functional studies are performed to determine if these are activating mutations or not.

A: We are sorry that we cannot include functional studies in the present manuscript. As we stated above we are planning to conduct experiments in near future.

Q: IN addition the third mutation is non-sense near the N-terminal of CSF3R and would be expected to create a non-functional protein. Again no functional studies to determine relevance of this "mutation". As tumor relevant CSF3R mutations cause activation of signaling, it is unclear what non-functional protein would have any role in leukemogenesis.

A: We agree to the reviewer about Q170X. As we said we did not do any functional studies around these mutations. We completely understand the concern of the reviewer and currently we cannot hypothesize the activation mechanism underlying Q170X mutant because it produces very short protein.

            Moreover, we do understand the mechanistic insights related to other frameshift or non-sense mutations in CSF3R that occur in the cytoplasmic domain of the protein (1). These nonsense mutations like Q741X, Y752X impair the receptor internalization and alters their interaction with other proteins SHP-1/2 and SOCS family members.

References

  1. Maxson JE, Gotlib J, Pollyea DA, Fleischman AG, Agarwal A, Eide CA, et al. Oncogenic CSF3R mutations in chronic neutrophilic leukemia and atypical CML. N Engl J Med. 2013;

Reviewer 3 Report

General comment: Rashid et al.; Used an Onco panel to investigate mutation in cancer
genes in 12 patients with B-ALL. My comments are below..
Introduction, Material and Methods
The introduction provides a good overview of issues relevant for the addressed studies.
However, I would like to suggest some improvements:
1. (line 62), please specify if those 12 patients were pre-treatment or not.. if
treatment was performed before the sample collection, each treatment received
for each patient has to be included.
2. Since you don´t have a normal counterpart for each patients, how did you
remove possible germline mutations?
3. (line 103). Why did you use the hg19 and not the hg38?
4. (line 106). The meaning of (MNV) was not mentioned before.
5. The word FISH/F.I.S.H was written in two different ways (line 69 and 255)
Results:
1. Is difficult to suggest that “CSF3R gene 28 including S661N, S557G, and Q170X
which might be associated with disease progression” when you just have one
time point for each patient;
2. The table was formatting issues; I would replace 15M for 15 months.. The gene
names are not in italic. Number of cells analyzed for patient (2112) is not
specify; I would replace failed for N/A or no mitosis. The column progression,
do not include much information.. (relapse when? How many days/months
after treatment; remission since when?)
Conclusion:
1. The paper end up focusing in the CSF3R gene alterations, I would at least do
some validation of this novel genetic variants using sanger sequencing..

Author Response

Q: General comment: Rashid et al.; Used an Onco panel to investigate mutation in cancer

genes in 12 patients with B-ALL. My comments are below..

Introduction, Material and Methods

The introduction provides a good overview of issues relevant for the addressed studies.

However, I would like to suggest some improvements:

  1. (line 62), please specify if those 12 patients were pre-treatment or not.. if

treatment was performed before the sample collection, each treatment received

for each patient has to be included.

A: The 12 B-ALL patients were not pre-treated. We collected the samples before treatment.

  1. Since you don´t have a normal counterpart for each patients, how did you

remove possible germline mutations?

A: Thank you for raising an important point. We did not sequence matching normal controls from the patients due to some reasons. But, we did try to obtain mutations that are enriched in somatic ones. In the processing step we filtered the genetic variants based on 1000 genome allele frequencies (AF). We removed those variants with AF greater than 0.1. In this process we lost about 10,000 variants that are supposed to be common polymorphism.

  1. (line 103). Why did you use the hg19 and not the hg38?

A: We used hg19 not for any specific reason. The Ion Proton sequencing platform was working on hg19 so produced the variant calling on hg19.

  1. (line 106). The meaning of (MNV) was not mentioned before.

A: MNV stands for multi-nucleotide variant. I mentioned the abbreviation in the manuscript at the first appearance at line number 106-107.

  1. The word FISH/F.I.S.H was written in two different ways (line 69 and 255)

A: We realized this error. We fixed this throughout the manuscript. We kept the term FISH to represent fluorescence in situ hybridization.

Results:

  1. Is difficult to suggest that “CSF3R gene 28 including S661N, S557G, and Q170X

which might be associated with disease progression” when you just have one

time point for each patient;

A: We understand the reviewer’s concern. We never claimed that these mutations are oncogenic mutations rather we said “these might be associated with disease progression”. In our opinion the causation will be claimed only after the functional validation.

  1. The table was formatting issues; I would replace 15M for 15 months.. The gene

names are not in italic. Number of cells analyzed for patient (2112) is not

specify; I would replace failed for N/A or no mitosis. The column progression,

do not include much information.. (relapse when? How many days/months

after treatment; remission since when?)

A: We are sorry for the formatting errors. We removed the ambiguities from the table in  the revised manuscript. 15M is not 15 months rather it says 15 years old male (M is for male). FISH results in the table are written as per ISCN (International System for human Cytogenetic nomenclature) guidelines. Those are not the gene names. We mentioned number of cells for patient 2112. Failed is replaced by NA.

            We are sorry we do not have the relapse/remission date for all the patients that are why we did not provide the relapse/remission dates.

Conclusion:

  1. The paper end up focusing in the CSF3R gene alterations, I would at least do

some validation of this novel genetic variants using sanger sequencing..

A: We agree with the reviewer. The circumstances and logistics did not allow us to do any validation experiments. We are sorry for that.

Round 2

Reviewer 3 Report

I am satisfied with the present format 

Author Response

We  thank you for showing your satisfaction with the present form of the manuscript.